# Craniofacial 3D Morphometric Analysis with Smartphone-Based Photogrammetry

**DOI:** 10.3390/s24010230

**Published:** 2023-12-30

**Authors:** Omar C. Quispe-Enriquez, Juan José Valero-Lanzuela, José Luis Lerma

**Affiliations:** Photogrammetry and Laser Scanner Research Group (GIFLE), Department of Cartographic Engineering, Geodesy and Photogrammetry, Universitat Politècnica de València, Camino de Vera s/n, 46022 Valencia, Spain; juavalan@topo.upv.es (J.J.V.-L.); jllerma@cgf.upv.es (J.L.L.)

**Keywords:** 3D model, cranial facial, smartphone device, photogrammetry, 3D scanning

## Abstract

Obtaining 3D craniofacial morphometric data is essential in a variety of medical and educational disciplines. In this study, we explore smartphone-based photogrammetry with photos and video recordings as an effective tool to create accurate and accessible metrics from head 3D models. The research involves the acquisition of craniofacial 3D models on both volunteers and head mannequins using a Samsung Galaxy S22 smartphone. For the photogrammetric processing, Agisoft Metashape v 1.7 and PhotoMeDAS software v 1.7 were used. The Academia 50 white-light scanner was used as reference data (ground truth). A comparison of the obtained 3D meshes was conducted, yielding the following results: 0.22 ± 1.29 mm for photogrammetry with camera photos, 0.47 ± 1.43 mm for videogrammetry with video frames, and 0.39 ± 1.02 mm for PhotoMeDAS. Similarly, anatomical points were measured and linear measurements extracted, yielding the following results: 0.75 mm for photogrammetry, 1 mm for videogrammetry, and 1.25 mm for PhotoMeDAS, despite large differences found in data acquisition and processing time among the four approaches. This study suggests the possibility of integrating photogrammetry either with photos or with video frames and the use of PhotoMeDAS to obtain overall craniofacial 3D models with significant applications in the medical fields of neurosurgery and maxillofacial surgery.

## 1. Introduction

Studying the shape of the human head and face is a multidisciplinary field of study of great importance in medicine, archaeology and anthropology. In the medical field and biomedical research, accuracy and efficiency in the evaluation of craniofacial morphology play an important role in various disciplines, from the identification of syndromic craniofacial disorders [1] to forensic anthropology [2,3]. The search for tools and techniques that allow the precise acquisition of 3D morphometric data has become a recurring objective in the scientific community.

Craniofacial morphometry, by obtaining three-dimensional models (3D), allows specialists to better identify alterations that could occur in alterations in the shape of the skull [4]. It is also recurrent to have the ability to identify craniosynostosis, which is the second most frequent type of craniofacial malformations [5]. Cases such as knowing the anatomy of the skull base of patients undergoing intrauterine or postnatal myelomeningocele repair allows doctors to determine its relationship with hydrocephalus [6]. It may also be the case to identify premature synostosis of the sutures to evaluate the relationship between craniofacial dysmorphology and suture pattern in children with plagiocephaly [7].

These cases are appropriate to indicate the usefulness of evaluating and monitoring [8], which allows the development of a better quality of care [9]. Accurate identification of anatomical points on the head and face is essential for the rigorous identification of key points, such as those located in the eye sockets, the shape of the nose, and mouth, among other points of interest. The identification of anatomical points is fundamental in studies that seek to quantify dysmorphic facial features [10] or the relationship between face symmetry [11] and the influence of diseases. Due to the importance of anatomical points, there is research exploring the detection of faces in digital images [12]. As well as the detection of cephalometric landmarks from digital images of frontal facial images in forensic medicine [13]. There are several ways to obtain 3D models of the head, each employing specific technologies and data capture methods. One of the best-known ways to acquire this information is the use of a 3D scanner, either laser or structured light; by using a specialised instrument, the user can record the 3D shape of the surface of the head by measuring the distance to reflected points. The advantage of using a 3D scanner is the benefit of overcoming the limitations of two-dimensional photographs while avoiding ionising radiation from computed tomography [14]. The metric quality of 3D scanners is constantly evaluated [15] with various settings when scanning the contour of the head [16,17,18].

Currently, with the advent of the 3D scanner, it has become very convenient for researchers to acquire precise 3D anthropometric measurements of the head and face [16]. There are alternatives to using the scanner to track volume changes without radiation exposure during treatment so that volume changes can be tracked in patients with head and neck cancer, thus avoiding the use of magnetic resonance imaging [17]. Photogrammetry is also a technique used to obtain 3D craniofacial models. In this process, multiple 2D images of the head and face are captured from different angles and used to reconstruct an accurate 3D model of the skull and face. Photogrammetry is based on triangulation, which involves measuring the distances and angles between known landmarks in 2D images to determine the 3D position of points in a derived 3D model, similar to 3D scanners.

Salahzadeh [18] shows that photogrammetry is a highly reliable method both between different evaluators and when the same evaluator applies it on different occasions. Orientation-calibrated stereophotogrammetry is presented as a highly accurate method to record the natural position of the head (NHP) [19]. In addition, photogrammetry allows the development of digitisation processes of the head, such as the rapid and precise location of electrodes and reference markers [20].

Smartphone-based photogrammetry offers key advantages, such as accessibility, portability, and high-resolution image capture, which are particularly beneficial in clinical and investigative settings where mobility and ease are crucial. Additional studies support the feasibility and effectiveness of smartphone photogrammetry in a wide variety of contexts [21,22,23], reinforcing our argument regarding its relevance in craniofacial morphology assessment. The widespread availability of smartphones can also cut costs, making this technique economically appealing. In comparison to laser and structured light scanners, smartphone photogrammetry can deliver comparable results in terms of accuracy and quality of the 3D models, as far as enough texture is available in the scene or artificial targets are placed on the object to achieve maximum accuracy. Although laser or 3D scanners have advantages in speed and the ability to handle complex surfaces, their limitations in terms of cost, size, and expertise are offset by the practicality of smartphone-based photogrammetry.

For a long time, obtaining high-quality and high-resolution 3D models, crucial for the identification of anatomical landmarks and detailed morphometric evaluations, was restricted to using professional cameras and specialised software, which often lay beyond the reach of healthcare professionals. However, with the rapid technological advancement in smartphones and the primary cameras of smartphones, it has become increasingly common to capture high-resolution and high-quality images. This now opens up the possibility of generating 3D models using smartphones [24,25,26], which can significantly contribute to cost reduction in processing overall.

Despite the availability of high-quality images, important questions arise when it comes to obtaining 3D models of individuals. The present study uses smartphones and their ability to capture high-quality images to offer an accessible, portable, and accurate solution for obtaining craniofacial morphology. Through smartphone-based photogrammetry, the results presented in this paper explore the creation of head and facial 3D models. Although subjects are instructed to keep their heads static, it is essential to investigate whether this can influence the generation of 3D models. This study aims to explore how the stability of head position during image capture using smartphones can impact the quality and accuracy of 3D models, with significant implications for its application in the fields of neurosurgery and maxillofacial surgery.

## 2. Materials and Methods

PhotoMeDAS (Version 1.7) is a photogrammetric tool used to analyse cranial deformation primarily focused on infants [20]. It was developed by the Photogrammetry and Laser Scanning Research Group (GIFLE) of the Department of Cartographic Engineering, Geodesy, and Photogrammetry at the Polytechnic University of Valencia (UPV).

### 2.1. PhotoMeDAS App

The procedure begins with the placement of a coded cap and three orientation stickers on the volunteer. Subsequently, the data is recorded using the mobile application PhotoMeDAS, Figure 1. Once this step is completed, the results obtained, along with the corresponding report, can be viewed on the PhotoMeDAS website [27].

### 2.2. Smartphone Data Acquisition (Photographs and Video)

The Samsung Galaxy S22 [28] is positioned as a standout choice due to its exceptional high-quality camera, capable of capturing high-resolution images and videos with autofocus, stabilisation, recording capabilities, and a fast processor. These features are detailed in Table 1. The selection of this smartphone addresses the need to provide an affordable alternative for monitoring head growth or deformation in individuals. Additionally, this device is compatible with the PhotoMeDAS mobile application in its Android version. The significance of the Samsung Galaxy S22 in the Android phone market was decisive in our choice, standing out as a reliable option to meet our specific needs.

### 2.3. 3D Scanner

The 3D data capture device Academia 50 [29], based on structured light technology, stands out for its ability to generate accurate and detailed 3D models of physical objects. This scanner is designed for use in various applications, such as reverse engineering, quality control, part inspection, and cultural heritage documentation (Figure 2).

### 2.4. Agisoft Metashape

Agisoft Metashape is a powerful image processing and photogrammetry tool that enables the creation of 3D models, orthophoto mosaics, and maps from images captured from the air or the ground [30]. Version 1.7 represents one of the latest updates released in October 2021 and stands out for its significant improvements in performance and stability (Figure 3a).

### 2.5. CloudCompare

CloudCompare v.2.13.alpha is an open-source software designed for 3D data visualisation, editing, and processing [31]. This software has facilitated the referencing process, calculating average distances and deviations between procedures. This includes comparing the model obtained with the Academia 50 scanner with models derived from photographs, video frames, and PhotoMeDAS (Figure 3b).

### 2.6. Volunteers

The participant selection was carried out through a call for volunteers who attended the facilities of the GIFLE research group at the UPV. Prioritising the diversity in head size, head shape and skin texture, ranging from children to adults, volunteers were from different regions (Europe and South America) with ages spanning from 3 to 28 years; males and females were included, plus two head mannequins with seamless colour shape replicating volunteers with cranial perimeters corresponding to one- and two-year-olds (Table 2). In addition to considering variability in physical characteristics, a specific assessment of participants’ behaviour and their ability to follow instructions during data collection was conducted. In fact, volunteers were instructed to maintain a fixed gaze to prevent head movements in order to optimise data collection. Each participant wore the PhotoMeDAS cap, accompanied by orientation stickers and round Academia 50 stickers.

## 3. Workflow

This study provides a detailed description of the methodology employed (Figure 4). A comparative assessment of craniofacial 3D models using smartphone-based photogrammetry was conducted. This methodology was applied to a total of 8 heads, 6 out of 8 humans and 2 mannequins.

To perform the procedure, a Samsung Galaxy S22 smartphone was used in conjunction with the PhotoMeDAS application, along with the Academia 50 white-light 3D scanner. Subsequently, Agisoft Metashape software was employed for photogrammetric processing. Using the reference model obtained with the 3D scanner, the referencing and scaling of the generated models were carried out.

Finally, model comparison was performed using Cloud Compare software to assess distances between the models, and v29.0 software was used for comparative statistical analysis of anatomical points around the heads.

### 3.1. Data Acquisition

Before commencing the scanning and data collection process with smartphones, volunteers were given detailed instructions regarding the overall procedure and the use of the PhotoMeDAS application. Subsequently, the coded cap and three necessary stickers for 3D scanning were used (Figure 5). PhotoMeDAS records the ArUco-coded targets to extract the four corner coordinates of each target. It is important to note that volunteers were instructed to maintain a stable position and focus their gaze on a fixed point on the wall during the data capture.

### 3.2. 3D Scanning Data Acquisition

The VXelement software was used to run the Academia 50 scanner [32]. Next, the 3D scanner was calibrated using a board with circular targets. Then, the configuration was set as follows: resolution of 0.5 mm, self-positioning with targets, geometry and texture, texture mapping, target filling, and precision in contour optimisation (Figure 6).

During the data acquisition, the 3D scanner was moved around the head, ensuring that all angles were covered and that the scanning object always remained in view. Upon completing the 3D scanning, a thorough review was conducted to ensure the integrity of the captured data. In case of any deficiencies, a new capture of the same model was performed. Once the 3D model was obtained, it was exported and stored for later import into another software for the referencing process. Worth noticing is that exporting usually took 10 to 20 min in addition to the data acquisition time.

### 3.3. Data Acquisition with the Smartphone in Camera and Video Modes

Data acquisition through the camera and video involved several considerations. Ensuring proper lighting and no obstructions between the model and the mobile device’s camera. Images were taken in “.jpg” format with default autofocus, ensuring that the volunteer’s head was in the centre of the frame and in constant focus.

Photographs were captured from multiple angles while maintaining a constant angle with the camera and keeping the volunteer’s head in view. Shots were taken at a fixed distance of approximately 20 to 25 cm from the head, with a 1-s interval between each shot. Two complete circles around the head and a partial one to include the chin were performed, along with a perpendicular set (Figure 7a).

For video recording, the same conditions were maintained, but the default video setting on the mobile was used, keeping automatic adjustments throughout the data capture process (Figure 7).

### 3.4. Data Acquisition with PhotoMeDAS

To carry out the measurement, it is essential to ensure that the entire visible surface of the cap is centred on the screen. A circle on the screen serves as a guide to visualise as many codes as possible. Additionally, maintaining a consistent distance when capturing head images is important for detecting targets. It is recommended to begin from the front, capturing the three stickers that serve as reference points. Once the application has collected sufficient data, it is sent to the server for processing [20]. After processing the data, the software generates reports and 3D models that are available on the PhotoMeDAS.eu website.

### 3.5. Data Treatment

A visual review of the photographs was conducted to ensure the quality of the data captured with the mobile camera, eliminating those that showed movement or had obstructions that hindered the proper visualisation of the volunteer’s head.

In the case of video recording, a Python script was implemented to break down the video into frames every 20 s. This technique simplified and expedited data processing. To achieve this, the “cv2” module was used, which corresponds to the widely recognised OpenCV library in the field of computer vision and enables effective work with images and videos (Figure 8).

### 3.6. Selection of Images for the Photogrammetric Processing

In this project, the total number of images indicated in Table 3 was used. To process these images, the quality of each one was evaluated, considering various data such as date, time, camera type, focal length, ISO, and capture speed, among others. Each image was analysed in terms of its quality, and formats were taken into account: the smartphone camera had a resolution of 4000 × 3000 pixels, while the frames extracted from the video had 1920 × 1080 pixels (Figure 9).

### 3.7. Photogrammetric and Videogrammetry Processing

The process includes camera calibration, photo orientation, creation of a dense point cloud, application of a confidence filter for data, selection of high-confidence points, generation of a three-dimensional mesh, and the addition of textures to create detailed and accurate 3D models from images. Each stage is crucial to ensure the precision and quality of the results (Figure 10).

The following parameters (Figure 11) in the photogrammetric process were selected to achieve the results. For the Align Photos in Agisoft Metashape (Figure 11a), the “Generic Preselection” technique was selected to determine the position and orientation of each photograph, yielding a sparse point cloud. Medium accuracy was employed, setting a maximum number of key points per photo and tie points up to one million. The final exterior orientation (photo alignment) is presented in Figure 12.

For the Dense point cloud (Figure 11b), the reuse depth maps option was selected, similar to “Calculate point colors” and “Calculate point confidence”. This latter option was considered crucial to filter out the points with low confidence in another step: confidence value >3.

### 3.8. Model Referencing

A reference coordinate system was set for each 3D model through Academia 50. To achieve this, anatomical reference points on the head, such as the vertices of the sticker, were identified and used as references in the Agisoft Metashape software to align the corresponding 3D models. This way, a common reference framework was established, allowing for precise and consistent comparison of the different 3D models obtained during the project.

With three reference points on each model, rotation and scaling of the models obtained through photogrammetry and video photogrammetry were performed. These reference points were distributed near each ear and at the front of the 3D model. Regarding the referencing of the model obtained with PhotoMeDAS, it was carried out in CloudCompare (Figure 13).

### 3.9. Visual Comparison of Meshing and Texturing

The results of the meshing and texturing of the head, which were obtained with photogrammetric processing and the 3D scanner, were compared. The figures correspond to the Front View (Figure 14 displays the results achieved with Academia 50 (Figure 14a) and Agisoft Metashape with camera photos (Figure 14b) and video frames (Figure 13c). A numerical comparison was conducted among the four models obtained, using the 3D scanner-generated model as a reference. For this purpose, CloudCompare tools were used, enabling the comparison of distances between point clouds and/or meshes. This way, precise measurements of the differences between the 3D models can be obtained, contributing to the evaluation of the quality and accuracy of the various acquisition methods. During this comparison, the points in the point cloud will be coloured based on their distances to the mesh using the C2M tool (Figure 15).

### 3.10. Anatomical Landmarks on the Head

Within the context of this research, a precise definition of craniofacial reference points or anatomical points on the head was established. Various markers such as predefined markers, skin marks, moles, face landmarks, and other clearly identifiable points on the textured 3D model were used. This methodology facilitated the comparison of the three-dimensional coordinates (x, y, z) of the points obtained through the different approaches.

After identifying the anatomical reference points of interest, 3D coordinates (X, Y, Z) were obtained for each evaluated area. This was achieved using Agisoft Metashape software, which allowed for proper 3D model referencing. Three points per area were considered, using mark vertices in some areas and face landmarks in others, such as the cap area and the face (Figure 16). This allowed for obtaining precise and detailed coordinates. In the context of PhotoMeDAS, the vertices and their respective 3D values were identified using the CloudCompare software.

## 4. Results

The 3D craniofacial models were successfully obtained from a variety of three smartphone-based photogrammetric and videogrammetry approaches and were compared with the cranial model obtained with the Academia 50 scanner. The quality of the models was assessed in terms of accuracy and level of detail of the represented anatomical structures.

### 4.1. Processing Time

A time comparison encompassing all stages of the process was conducted (Table 4). It is important to note that in no case was the time for fitting the cap and markers included. Specifically, the time from the recognition of coded stickers to obtaining the 3D model on https://photomedas.eu (accessed on 11 October 2023) was considered for the PhotoMeDAS case. For the camera and video, the time dedicated to image acquisition and photogrammetric and videogrammetry processing in Agisoft Metashape was assessed. Regarding the 3D scanner, the time spent on data capture and the 3D model export process were analysed (cf. Figure 5).

### 4.2. Comparison of 3D Mesh

Table 5 summarises the general statistics after comparing the distances (bias) between the 3D meshes obtained through different acquisition approaches: photogrammetry, videogrammetry and PhotoMeDAS. The comparison was executed in CloudCompare, reporting the mean and standard deviation values for each instrument used, considering as a reference the meshing obtained with the 3D scanner.

Figure 17 graphically represents the values indicated in Table 5. The comparison is divided by model and approach (cf. Figure 5) as follows: (C) Photogrammetry with camera, (V) Videogrammetry with video, and (P) PhotoMeDAS.

### 4.3. Number of Faces of the 3D Models

The number of faces in a 3D model refers to the polygons that make up its surface. These faces are typically triangular or quadrilateral and are responsible for defining the shape and contours of the 3D object. As the number of faces in a 3D model increases, so does its level of detail and complexity.

In Figure 18, the results of the procedures are presented comparatively. However, it is evident that due to the non-uniformity in data capture, it becomes necessary to undertake a standardisation process for a precise comparison in subsequent stages.

To standardise the comparison, the number of faces required to build the mesh covering the coded cap was quantified (Figure 19). The results obtained in the comparisons show that the 3D model obtained through the 3D scanner exhibits higher quality in terms of the 3D representation of the object (Table 6). On the other hand, the models obtained through the camera and video show lower resolution in comparison with the Academia 50 scanner. Additionally, the 3D model obtained with PhotoMeDAS has the lowest number of faces in the mesh compared to the other models, indicating potentially lower precision in representing finer object details.

### 4.4. Anatomical Reference Points

For the process of identifying anatomical points, the texture of the 3D model is important; it refers to how the surface of an object or character is perceived and felt in a 3D environment. This is achieved by applying images or patterns to the model’s surface to simulate visual details such as colour, texture, and materials. Textures are crucial for adding realism and detail to objects in a 3D world, allowing for the representation of everything from the smoothness of skin to the roughness of a rock.

In the texturing process, textures are carefully adjusted and mapped onto the coordinates of the 3D model to achieve a convincing and realistic final representation when rendering the scene. In Figure 20, you can see the results obtained using 3D scanning, photogrammetry, and videogrammetry.

Figure 21 illustrates the manual extraction of the three-dimensional (3D) coordinates related to the different targets. This process is carried out using the textured model derived from photogrammetry and videogrammetry processing, as well as the texturised 3D model obtained with the 3D scanner. In the case of PhotoMeDAS, the coordinates of the vertices of the coded stickers are utilised.

In the comparison of anatomical reference points (Figure 21), the following procedure was followed: three coordinates (x, y, z) were obtained for each area of the head (Figure 15), assuming a local coordinate system defined by the 3D scanner. Subsequently, the distance between the identified point in the textured model of the scanner and its counterpart in the textured model of the camera and video was determined. In the case of PhotoMeDAS, the coordinates of the vertices of the coded cap were used as an anatomical reference. It is worth noting that, due to the absence of a specific area for the face in the PhotoMeDAS version, this area was not considered for evaluation.

The evaluation of anatomical reference points is presented in Table 7. From this table, it can be observed that the greatest variation occurs in the case of volunteer V 2. Excluding the case of V 2, the other values are more uniform, averaging around 1.5 mm.

Figure 22 shows a summary of the mean distance (bias) using the anatomical reference points by photogrammetric approach versus Academia 50. It should be noted that the dataset corresponding to Volunteer 2 (V 2) exhibits the largest differences due to uncontrolled patient movement during data acquisition.

### 4.5. Student’s T and Anatomical Reference Points

To conduct the t-student test, 3D coordinates (x, y, z) of anatomical reference points on different head areas were used. These 3D points were identified through the texturing of the model except for PhotoMeDAS due to the availability of the 3D coordinates.

Hypothesis Formulation
**Null** **Hypothesis** **(H0):**States that there is no significant difference (bias) between the means of the two groups. The test was conducted for the average distance of each case (photogrammetry (Camera), videogrammetry (Video), and PhotoMeDAS) compared to the 3D scanner, being assessed for a bias of 0 mm (no bias), and 0.25 mm, 0.5 mm, 0.75 mm, 1 mm, and 1.25 mm.
**Alternative** **Hypothesis** **(H1):**States that there is a significant difference between the means of the two groups. The sample mean is not close to the value of the H0 hypothesis.

To achieve the objective and determine equivalence, the IBM-SPSS Statistics software was used, using a one-sample T-test, considering the ideal separation distance between models as the reference. Table 8 shows the results achieved with a 95% confidence level, according to the approach.

For this evaluation, the case of volunteer V 2 was not considered, as this individual exhibited more movement than all other cases, distorting the distribution of the statistics.

If the significance level (*p*-value) is greater than alpha (0.05), the null hypothesis H0 is not rejected. If the significance level is less than alpha (0.05), the null hypothesis H0 is rejected.

It is determined that the average distance (bias) of the photogrammetry and videogrammetry procedures is equivalent (there is no significant difference) to the 3D scanning results, using photogrammetry with the smartphone camera up to 0.75 mm, for videogrammetry up to 1 mm, and for PhotoMeDAS up to 1.25 mm (Table 8).

## 5. Discussion

In all the solutions presented in the research presented in this paper, one single imaging sensor/3D scanner was used for data collection. In addition, targets were considered a premise of this experimentation. Thirty-two 3D models were obtained through various approaches: eight photogrammetry models, eight videogrammetry models, eight models generated with PhotoMeDAS, and eight 3D scanning models. The latter was used as the reference 3D model (i.e., ground truth) for the automatic comparison of meshes and linear distances between anatomical landmarks. This achievement was made possible thanks to the participation of eight volunteers, whose ages ranged from 1 to 28 years, as indicated in Table 2.

In both the photogrammetry and videogrammetry processes, data capture was carried out around the head following the guidelines illustrated in Figure 7a. However, in general terms, a lower number of images were used in the videogrammetry process, with an average of 111 frames, compared to the photogrammetry process, which had an average of 226 images. In the case of videogrammetry, priority was given to ensuring uniform movement around the head, thanks to a built-in autofocus system in the video that ensured the capture of stable and noise-free frames. The images used were obtained through a processing in which the extraction of one image was configured for every 20 video frames (Figure 8). On the other hand, in the case of photogrammetry, due to the lack of precision in the overlapping areas when taking photographs, there was some redundancy in data acquisition in some cases.

Worth noticing is that imagery obtained with the camera (4000 × 3000) for the photogrammetric processing is of higher resolution than the frames obtained from the video (1080 × 1920) for the videogrammetry processing. This means that camera images can register more details than video frame counterparts. However, video frames have an important advantage: they can capture images at higher speed, minimising uncontrolled volunteer motion. This is especially useful in situations where the user wants to capture the dynamics of a moving object or person. Slow motion with a higher frame rate of 60 Hz helps to overcome this uncontrolled situation (cf. results presented in Table 5 and Figure 16, V 2).

As expected, the white-lite 3D scanner is an ideal approach for obtaining high-quality head 3D models due to its high accuracy, precision, and speed capture in real time. Nevertheless, it is usually an expensive approach, and whenever budget is a limiting factor, the smartphone-based camera and video might serve as viable alternatives, depending on the application.

The statistical results presented in Figure 17 and Figure 22 reveal the notable influence of head movement on data collection procedures in photogrammetry, videogrammetry and 3D scanning. Thanks to the inclusion of stickers, reliable results were almost always obtained. Previous experiences before the presented research allow us to confirm that the deliverables were non-metrically reliable and useless. In the specific case of sample V2, consisting of a 6-year-old child, despite instructions to keep the head static and fixed on a designated point, involuntary movements were observed during the data acquisition phase. Although the 3D model obtained through photogrammetry and videogrammetry provided a considerable level of detail for texturing the model, its susceptibility to movement is significant. Conversely, within the context of the Academia 50 sample, it is evident that the use of a substantial number of circular stickers allows for movement tolerance without compromising the quality or geometric coherence of the generated craniofacial 3D model. Additionally, in the case of PhotoMeDAS, the capability of its data capture system relies on the precise recognition of coded stickers on the head region, granting it the ability to tolerate the child’s movements during interaction with the mobile device.

A higher amount of noise and morphological variation can be observed in the delivered photogrammetry models obtained through the camera and video approaches (Figure 18b,c), especially in the areas not covered by the cap used during data acquisition. To minimise interference in the photogrammetric approach, applying the confidence filter before meshing is of utmost importance (cf. Figure 10). Whereas the 3D scanner mesh is highly smooth and displays the finer details, such as the seam and the shape of the coded targets, the photogrammetry approach also displays the seam, but there is a higher level of mesh roughness across the whole head (Figure 18b). Even noisier is the videogrammetry approach, where the seam can be detected, but the lack of smoothness is really apparent (Figure 18c). The lowest resolution is clearly achieved with the PhotoMeDAS approach due to the fact that its final aim is not to extract the best shape but to compute up to a millimetre head shape (Figure 18d) from which to deliver autonomous cranial anthropometric reports. Nevertheless, this latter approach automatically delivers the head’s shape and deformation in comparison with an ideal patient’s head, making this information useful for paediatricians and neurosurgeons monitoring cranial growth.

In the process of identifying anatomical points, it is crucial to consider the texture resolution, as it defines and facilitates the identification of features such as marks, moles, blemishes, sticker vertices, and other distinctive details that may be present on the head. In this context, it is specified that the resolution of Academia 50 varies between 50 and 150 DPI (i.e., equivalent GSD of 0.51–0.17 mm/pixel), photogrammetry has an average resolution of 0.07 mm/pixel, and videogrammetry has an average resolution of 0.2 mm/pixel. This facilitates precise identification of the vertices of the coded marks and exact referencing, with the smallest unit of the target being approximately 1 mm. Being able to clearly identify reference points with the support of texturing helps users to minimise errors in the referencing process (Section 3). Nevertheless, the quality of texture colours is substantially higher with photogrammetry or videogrammetry in comparison with the 3D scanning counterpart (cd. Figure 20a in comparison with Figure 20b,c).

Currently, on the market, there are comprehensive solutions for obtaining 3D models of the head that are tolerant to movement and efficient in data capture. These solutions include multi-camera or multi-scanner 3D systems, such as the 3dMDface™ and 3dMDtrio™ systems. Despite their effectiveness, these solutions can be expensive [33] for a large number of health institutions. When analysed individually, errors tend to be close to 0.35 mm, with a standard deviation of 0.14 mm, but comparative errors of almost 1 mm are achieved when comparing faces between different instruments [18].

The results presented as part of the research undertaken herein indicate that when a 3D model with high mesh resolution and texturing is required to perform an accurate morphometric analysis, a photogrammetric smartphone can achieve a comparative 3D model of differences 0.22 ± 1.29 mm (Table 5). The results obtained in this study are consistent with the prior research conducted in the work [26], which examined the acquisition of head measurements using two different approaches. On one hand, the traditional manual measurement technique was employed, and as an alternative, videogrammetry was used. For videogrammetry, reference marks were placed on the cap, and three measurements were obtained manually for the scaling of the 3D model. The comparison in that study revealed differences of 2 mm with a standard deviation of ± 0.9 mm [34]. This option is particularly suitable for measuring volunteers able to keep their heads still for at least 2 min. Furthermore, it can also be applied in the field of forensic examination [3].

In circumstances where expeditious data acquisition is required, typically within a timeframe of approximately one to two minutes, it might be interesting to contemplate the utilisation of videogrammetry. This alternative yields a comparative 3D model of differences of 0.47 ± 1.43 mm (Table 5). However, it is imperative to point out that when compared to photogrammetry, the resultant mesh will exhibit reduced details, although the texturing will remain akin. This procedure becomes beneficial when capturing data from specific regions of the head, such as the facial area, as keeping the face or head immobilised for extended periods can be challenging. Additionally, it can be harnessed for the analysis of cranial deformation using the smartphone video option [35]. This methodology has yielded the assessment of parameters regarding the differences in the semi-axes of adjusted ellipsoids, with a maximum deviation of 1 mm.

In situations where volunteers, especially infants and babies, are unable to keep their heads still, and there is a need for rapid acquisition of a 3D model of the skull, resorting to PhotoMeDAS proves highly advantageous. This argument is supported by the results obtained in the case of V 2, a six-year-old child who performed involuntary head movements during data acquisition. According to the findings, PhotoMeDAS delivered the most secure comparative performance, whether assessing the mesh model (Figure 17) or considering the anatomical reference points (Figure 22). This solution yields a comparative 3D model result of 0.39 ± 1.02 mm (Table 5), being particularly well-suited to accommodate scenarios with uncontrolled head movements, thanks to the use of the cap.

The accuracy results in measuring landmarks (Table 7) are in line with other smartphone-based photogrammetric studies independently of the model and branch of smartphones used [36,37,38]. In [36], the authors point out the significant differences between the individual camera models in terms of interior orientation stability and propose using geometric pre-calibration beforehand instead of self-calibration. However, the authors in [37] emphasise the need for proper scaling as being a more relevant factor for improving smartphone-based accuracy rather than performing geometric calibration in a separate process or pre-calibration. In addition, ref. [38] reports on the relevance of using the highest resolution smartphone cameras to provide the lowest localisation errors.

Regarding data capture, photogrammetry and videogrammetry (passive technology) were sensitive to light indoors, whereas the 3D scanner (active technology) experienced no issues in the laboratory environment, being able to create the craniofacial 3D model with lights turned off. Homogenous tungsten indoor light was used to record data. However, there was no chance to test the three solutions outdoors, where additional issues might have appeared with direct sunlight.

A data acquisition note is that images were captured with autofocus on instead of conventionally agreed autofocus off for photogrammetric applications. The final quality of the images was substantially higher whenever the autofocus was activated, and the final photogrammetric results did not decrease the parameter estimates. Also worth mentioning is the mandatory use of a cap with targets to minimise the slight and/or unexpected volunteer movement. With infants and kids, solving this point is a limiting factor to success because it is hard to get the ideal image network geometry surrounding the head to extract craniofacial 3D data during conventional consultancy using a single imaging sensor/3D scanner.

## 6. Conclusions

This paper has checked the performance of three smartphone-based photogrammetric approaches for craniofacial 3D morphometric analysis. The study concludes that the head 3D models obtained through different procedures, photogrammetry with camera acquisition, videogrammetry with video recording, PhotoMeDAS, and 3D white-light scanning (considered as reference data) have a significant impact on the metrics, processing time, visual quality and level of detail of the derived 3D models. The 3D while-light scanning approach with Academia 50 produced the highest average number of faces, while smartphone-based photogrammetry and videogrammetry exhibited noisier deliverables, namely in areas not covered by the cap during data acquisition, emphasising the need to apply reliability filters to remove gross errors and achieve better meshing and texturing results.

In the comparative evaluation of the 3D model and the linear landmarks, the three smartphone-based photogrammetric solutions yielded equivalent results despite proceeding times varying substantially (from mean values of 7 min for PhotoMeDAS, 12 min for videogrammetry and 103 min for photogrammetry), although acquisition time was similar and always better than 5 min. Worth noticing is that the usage of the white-light 3D scanner required equivalent data acquisition times (average 3 min) without ambient light but extensive processing time (35 min). In addition, PhotoMeDAS only records targets, so there was no chance to measure uncoded facial zones.

Regarding the comparison of linear characteristics with anatomical reference points, a statistical analysis was performed to evaluate the equivalence of distances between the white-light 3D scanner Academia 50 and the other methods, yielding equivalent statistical results of 0.75 mm for photogrammetry, 1 mm for videogrammetry and 1.25 mm for PhotoMeDAS.

The utilisation of smartphone-based photogrammetry opens the door to present applications in the healthcare domain. Its capacity to generate precise craniofacial 3D models enables a detailed and personalised diagnosis that might be particularly valuable in reconstructive surgeries and aesthetic procedures. Furthermore, it facilitates the geometric monitoring of treatments and the assessment of malformations, promoting continuous tracking of facial disease progression. In the field of telemedicine, smartphone-based photogrammetry can also contribute to innovating remote collaboration among healthcare professionals, enhancing coordination in patient diagnosis and treatment. Additionally, it might contribute to clinical research by providing crucial data for the development of new procedures and the enhancement of existing techniques in craniofacial surgery, solidifying its role as a transformative technology in medical practice.

In the future, the authors will research the influence of using smartphones coming from different manufacturers and with various features (including range-based technology) and the behaviour of lighting.

## Figures and Tables

**Figure 1 sensors-24-00230-f001:**
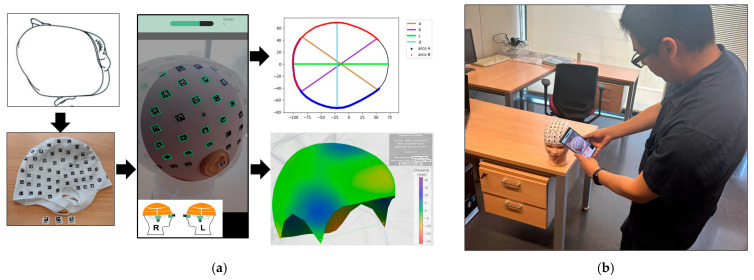
Materials: (**a**) PhotoMeDAS working system; (**b**) PhotoMeDAS data capture.

**Figure 2 sensors-24-00230-f002:**
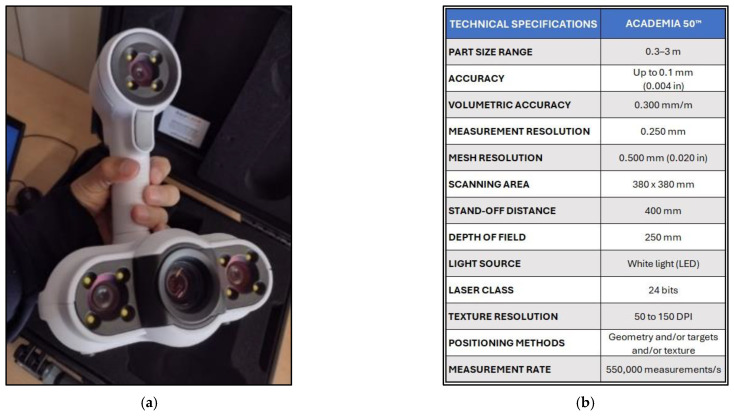
Academia 50 3D scanner: (**a**) instrument; (**b**) technical specifications taken from https://www.creaform3d.com/ (accessed on 11 October 2023).

**Figure 3 sensors-24-00230-f003:**
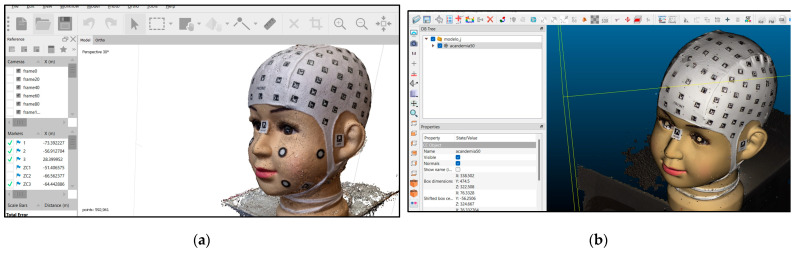
3D data: (**a**) Point cloud visualisation in Agisoft Metashape; (**b**) 3D model obtained with Academia 50 and imported into CloudCompare.

**Figure 4 sensors-24-00230-f004:**
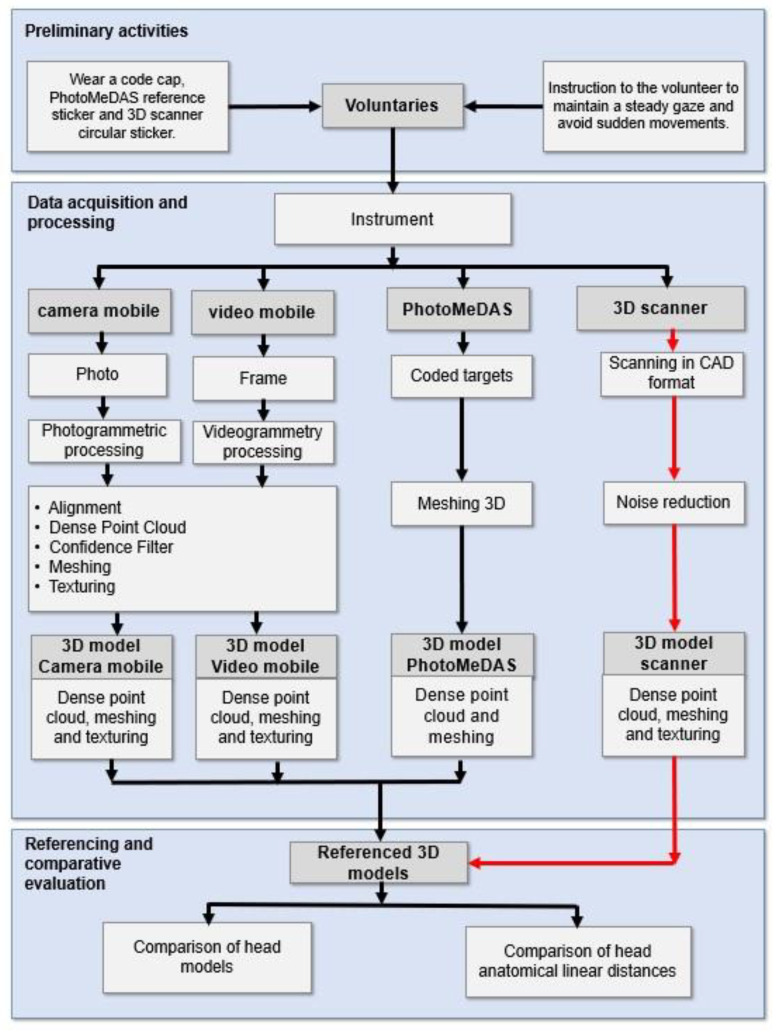
Workflow schema. The red arrow represents the reference model workflow, and the black arrow represents the workflow under evaluation.

**Figure 5 sensors-24-00230-f005:**
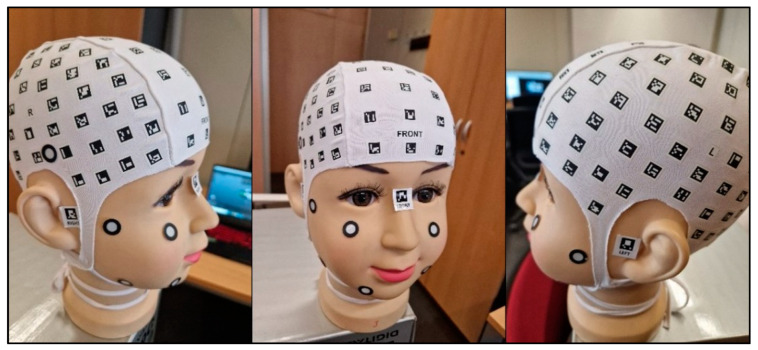
Coded cap and three stickers for PhotoMeDAS and circular stickers for the 3D scanner.

**Figure 6 sensors-24-00230-f006:**
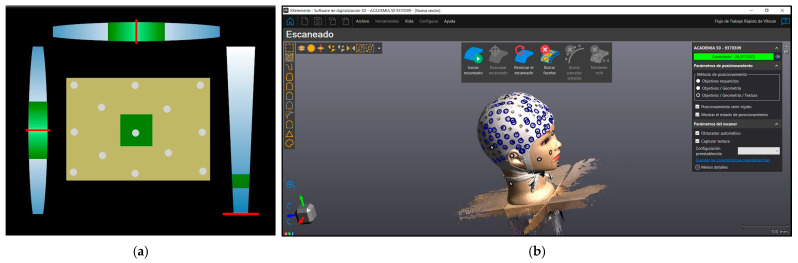
(**a**) Calibration processing in VXelement software v. 8.0.0, for the scanner calibration process, align the red lines with the green boxes; (**b**) Screenshot during the scanning data acquisition.

**Figure 7 sensors-24-00230-f007:**
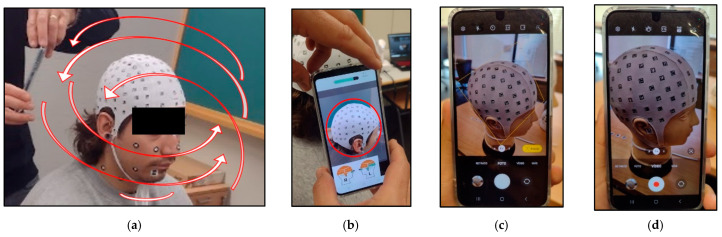
Galaxy S22: (**a**) Data acquisition strips, the red arrows represent the planned path using the smartphone; (**b**) PhotoMeDAS app, the red circle represents the focus area used by the app for sticker identification; (**c**) Camera mode, the square represents the smartphone’s automatic focus; (**d**) Video mode.

**Figure 8 sensors-24-00230-f008:**
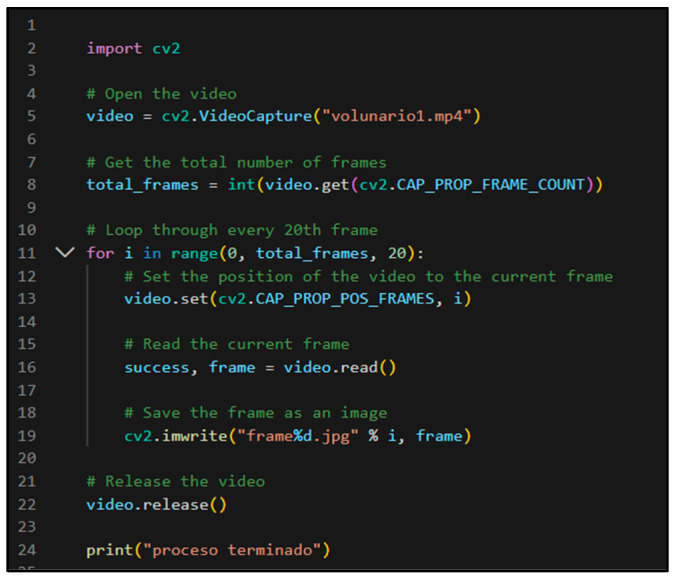
Code for converting video to frame.

**Figure 9 sensors-24-00230-f009:**
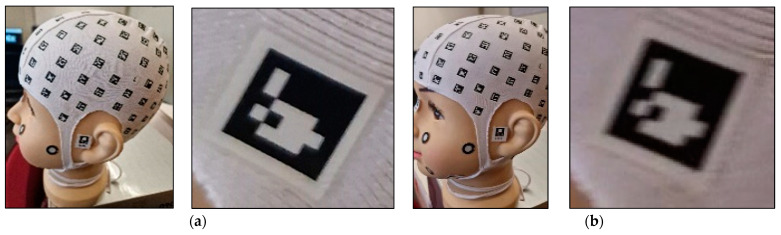
Comparison of input images: (**a**) Zoom in of an image obtained with the camera (4000 × 3000 pix); (**b**) Zoom in of an image obtained with the video (1080 × 1920 pix).

**Figure 10 sensors-24-00230-f010:**
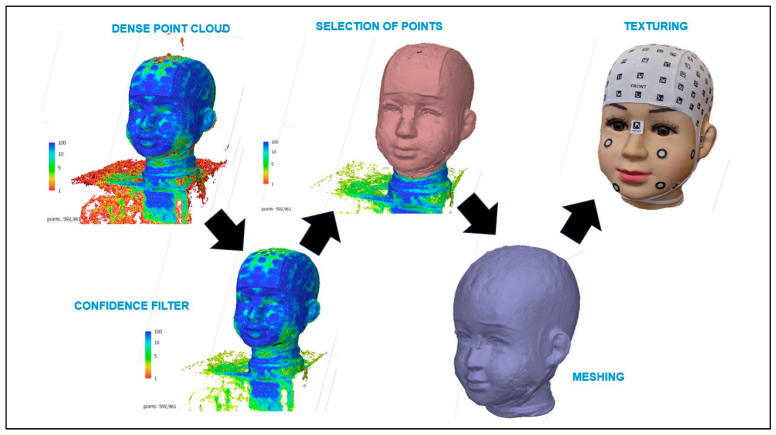
Photogrammetric and videogrammetry processing.

**Figure 11 sensors-24-00230-f011:**
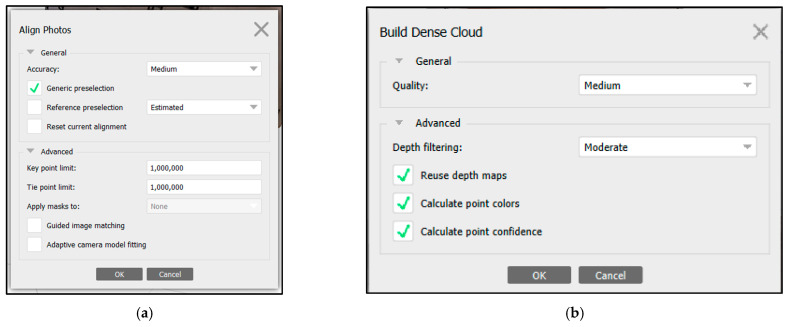
Set up in Agisoft Metashape: (**a**) Photo Alignment; (**b**) Dense Point Cloud Generation.

**Figure 12 sensors-24-00230-f012:**
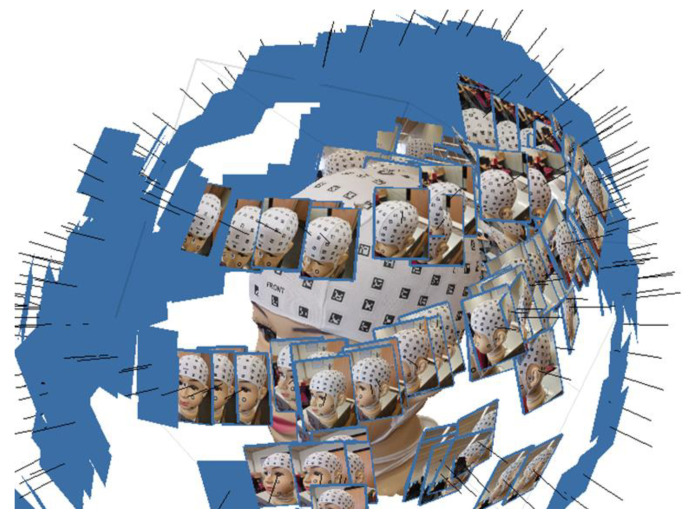
Final photogrammetric exterior orientation (alignment) used to create the 3D model presented in Figure 10. The black lines represent the camera’s projection center, and the blue areas represent the photography coverage area.

**Figure 13 sensors-24-00230-f013:**
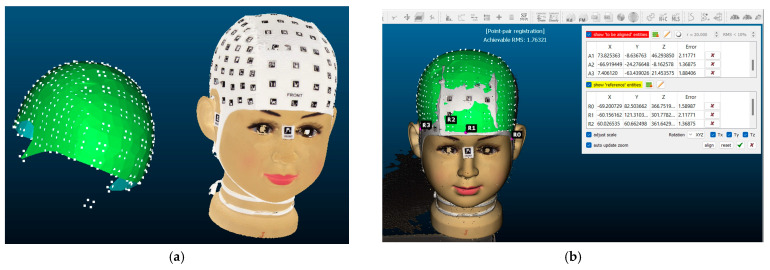
PhotoMeDAS model referencing: (**a**) PhotoMeDAS 3D model before referencing; (**b**) referenced model. The term ‘reference entities’ refers to the main model with its coordinate system, whereas ‘entities to be aligned’ refer to the model that will be adjusted to match the primary reference system. A scaling factor is used to adjust the model to the dimensions of the reference model, and the RMS error represents the discrepancy between them. The scaling factor is employed to standardise the 3D models and achieve homogenisation.

**Figure 14 sensors-24-00230-f014:**
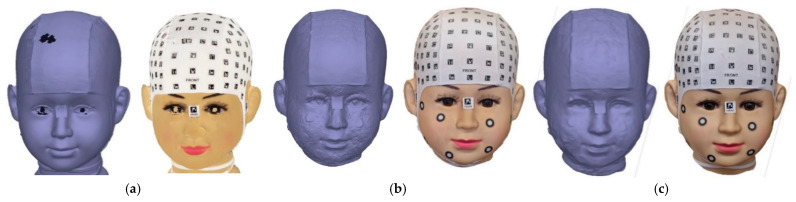
Front view after meshing and texturing:(**a**) Academia 50 scanner; (**b**) Photogrammetry; (**c**) Videogrammetry.

**Figure 15 sensors-24-00230-f015:**

C2M results among 3D models: (**a**) Scanner/Photogrammetry; (**b**) Scanner/Videogrammetry; (**c**) Scanner/PhotoMeDAS.

**Figure 16 sensors-24-00230-f016:**
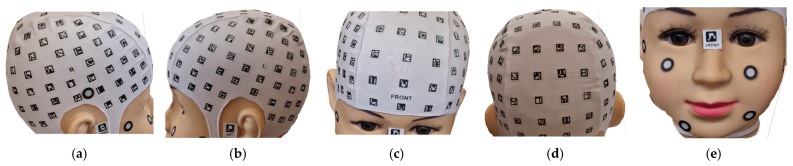
Zoning of the head: (**a**) Right Side Parietal Zone, ZPR; (**b**) Left Side Parietal Zone, ZPL; (**c**) Frontal Zone, ZF; (**d**) Posterior Zone, ZP; (**e**) Face Zone, ZF.

**Figure 17 sensors-24-00230-f017:**
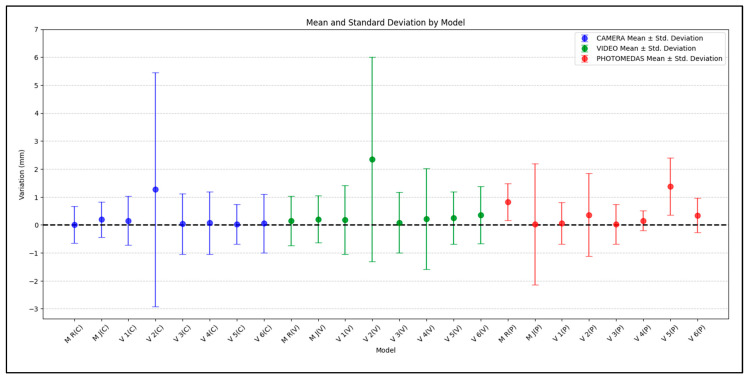
Average distance bias (dots) and ranges (+/− 1σ) with respect to the Academia 50 models.

**Figure 18 sensors-24-00230-f018:**
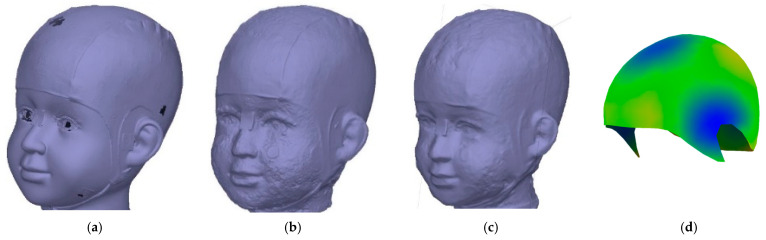
Comparison—Model J: (**a**) Academia 50; (**b**) Photogrammetry; (**c**) Videogrammetry; (**d**) PhotoMeDAS highlighting the deformation with an ideal head.

**Figure 19 sensors-24-00230-f019:**
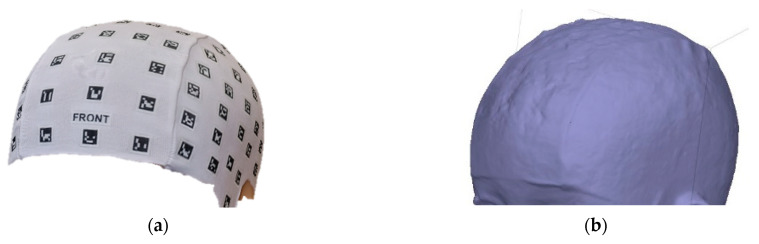
Comparison areas: (**a**) Identification of the area covered by the coded cap; (**b**) Area delineation for comparing approaches.

**Figure 20 sensors-24-00230-f020:**
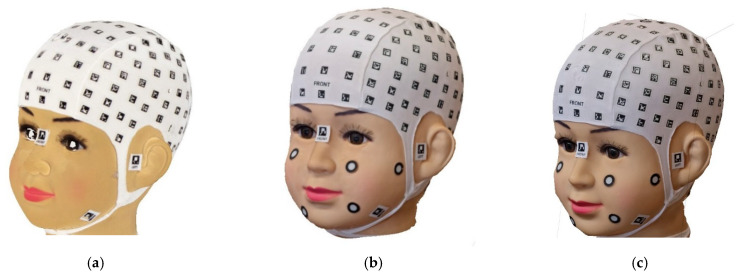
Texturing comparison—Model J: (**a**) Academia 50; (**b**) Photogrammetry; (**c**) Videogrammetry.

**Figure 21 sensors-24-00230-f021:**
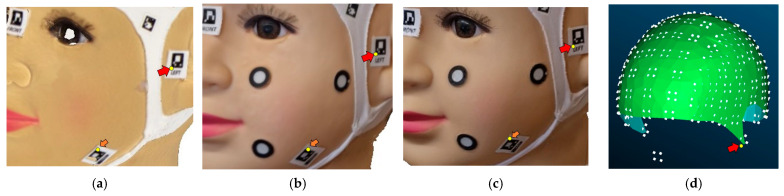
Extracted target coordinates from the 3D models: (**a**) Academia 50; (**b**) Texturised model from the camera; (**c**) Texturised model from the video; (**d**) PhotoMeDAS coordinates. The yellow point represents an anatomical point in the 3D models. The red arrow indicates an example of an anatomical point in the skull, and the orange arrow indicates a facial anatomical point.

**Figure 22 sensors-24-00230-f022:**
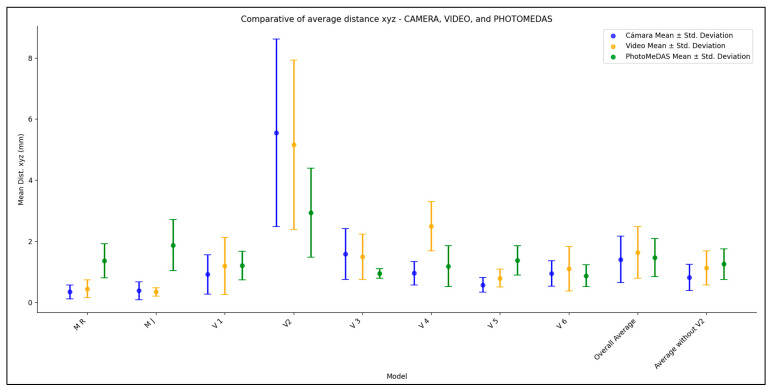
Mean distances on anatomical head reference points between models.

**Table 1 sensors-24-00230-t001:** Samsung Galaxy S22 Specifications.

Characteristic	Description
System	Operating System Android 12, Processor Exynos 2200 Octa-Core
Connectivity	Mobile Network 5G, WIFI 802.11 a/b/g/n/ac, Bluetooth v5.0, NFC Yes
Display	Size 6.1″, Resolution 2340 × 1080 px.
Camera	For photogrammetry, 4000 × 3000 pixels; for videogrammetry, 1080 × 1920 pixels
Memory	Internal 128 GB, RAM 8 GB

Source: https://www.samsung.com/es/smartphones/galaxy-s22/models/ (accessed on 11 October 2023).

**Table 2 sensors-24-00230-t002:** Data acquisition timespan and volunteer’s age.

Volunteer	3D Scanner	Camera	Video	PhotoMeDAS	Age	Description
Model R (M R)	3 min 58 s	2 min 13 s	1 min 18 s	4 min	1 year	Head mannequin
Model J (M J)	4 min 20 s	2 min 01 s	1 min 30 s	4 min	2 years	Head mannequin
Volunteer 1 (V 1)	1 min 55 s	1 min 12 s	1 min 10 s	5 min	3 years	Female
Volunteer 2 (V 2)	2 min 03 s	1 min 50 s	1 min 30 s	5 min	6 years	Male
Volunteer 3 (V 3)	2 min 36 s	1 min 50 s	1 min 07 s	5 min	14 years	Male
Volunteer 4 (V 4)	5 min 00 s	2 min 30 s	1 min 39 s	5 min	25 years	Male
Volunteer 5 (V 5)	2 min 57 s	2 min 02 s	1 min 20 s	5 min	27 years	Female
Volunteer 6 (V 6)	3 min 40 s	2 min 59 s	2 min 09 s	5 min	28 years	Male

**Table 3 sensors-24-00230-t003:** Summary of the number of images used to generate 3D models.

Model	M R	M J	V 1	V 2	V 3	V 4	V5	V6
Photogrammetry	189	283	261	220	212	225	261	155
Videogrammetry	139	118	66	102	108	138	108	175

**Table 4 sensors-24-00230-t004:** Overall processing time with different approaches.

Model	PhotoMeDAS	Photogrammetry	Videogrammetry	Academia 50 Scanner
M R	3.5 min	76.9 min	11.4 min	(30–40) min
M J	3.0 min	71.3 min	10.2 min	(30–40) min
V 1	8.2 min	64.9 min	16.2 min	(30–40) min
V 2	12.2 min	97.6 min	9.2 min	(30–40) min
V 3	11.1 min	75.3 min	8.1 min	(30–40) min
V 4	3.3 min	115.1 min	13.6 min	(30–40) min
V 5	7.2 min	235.9 min	10.0 min	(30–40) min
V 6	4.8 min	86.2 min	17.3 min	(30–40) min

**Table 5 sensors-24-00230-t005:** Distances between the meshes (Photogrammetry, Videogrammetry, and PhotoMeDAS) and the 3D scanner mesh.

Volunteer	Photogrammetry		Videogrammetry		PhotoMeDAS	
	$\bar{x}$ (mm)	σ (mm)	$\bar{x}$ (mm)	σ (mm)	$\bar{x}$ (mm)	σ (mm)
M R	0.00	0.66	0.14	0.88	0.82	0.66
M J	0.19	0.63	0.20	0.84	0.02	**2.17**
V 1	0.15	0.87	0.18	1.26	0.06	0.75
V 2	**1.26**	**4.18**	**2.34**	**3.66**	0.36	1.48
V 3	0.04	1.09	0.08	1.09	0.02	0.71
V 4	0.07	1.12	0.21	1.80	0.15	0.36
V 5	0.02	0.71	0.25	0.94	**1.38**	1.02
V 6	0.05	1.05	0.36	1.02	0.34	0.62
Average	0.22	1.29	0.47	1.44	0.39	0.97

The bolded text represents the maximum value when comparing the methods and volunteer models.

**Table 6 sensors-24-00230-t006:** Comparison of the number of faces in the meshing of the four approaches.

Volunteer	Academia 50	Photogrammetry	Videogrammetry	PhotoMeDAS
M R	301,986	391,965	111,849	1038
M J	332,552	461,530	98,119	1045
V 1	392,642	565,065	111,672	1031
V 2	420,262	588,931	98,745	1041
V 3	408,642	319,257	75,626	1030
V 4	460,980	447,549	79,580	1041
V 5	429,127	650,590	82,852	1047
V 6	411,820	278,301	60,027	1039

**Table 7 sensors-24-00230-t007:** Accuracy results for the three approaches.

Volunteer	Photogrammetry		Videogrammetry		PhotoMeDAS	
	$\bar{x}$ (mm)	σ (mm)	$\bar{x}$ (mm)	σ (mm)	$\bar{x}$ (mm)	σ (mm)
M R	0.35	0.23	0.45	0.29	1.37	0.56
M J	0.39	0.29	0.35	0.14	1.88	0.84
V 1	0.92	0.64	1.20	0.94	1.21	0.47
V 2	**5.55**	**3.07**	**5.16**	**2.77**	**2.94**	**1.45**
V 3	1.59	0.83	1.50	0.74	0.95	0.16
V 4	0.96	0.38	2.50	0.81	1.19	0.67
V 5	0.58	0.24	0.80	0.29	1.38	0.48
V 6	0.95	0.42	1.11	0.73	0.88	0.36
Average	1.41	0.76	1.64	0.84	1.47	0.62
Average without V 2	0.82	0.43	1.13	0.56	1.26	0.50

The bolded text represents the maximum value when comparing the methods and volunteer models.

**Table 8 sensors-24-00230-t008:** 2-tailed Student’s T *p*-values summary for the different approaches.

Approaches	0 mm	0.25 mm	0.5 mm	0.75 mm	1 mm	1.25 mm
Photogrammetry	0.000	0.000	<0.001	0.238	-	-
Videogrammetry	0.000	0.000	<0.001	0.000	0.142	-
PhotoMeDAS	0.000	0.000	<0.001	0.000	0.000	0.827

## Data Availability

The data presented in this study are available on request from the group leader, Prof. José Luis Lerma.

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
