# Peer review of "Craniofacial 3D Morphometric Analysis with Smartphone-Based Photogrammetry"

_sensors, 2023, doi:10.3390/s24010230_

Round 1

Reviewer 1 Report

Comments and Suggestions for Authors

It seems that this manuscript only introduces and follows the commercial or open-source software to handle the 3D facial model, including 3D scanner and photogrammetric. Mainly sections only show the setups of the software, without any contribution or improvement of the existing methods. Frankly speaking, I can not find the meaning of this research. Also, I am not sure it is a good way to use markers on a human face, especially in medicine. As I know, many 3D scanners and 3D reconstructing methods do not need markers.

At least, I hope the authors can discuss the non-rigid or moving conditions of the volunteers, i.e., how to robustly deak the face moving or expression change when taking the video or images, which often impact the real applications.

In all, I am inclined to reject this manuscript in the next round unless the authors can convincingly respond to my above issues.

Reviewer 2 Report

Comments and Suggestions for Authors

The topic of the article is very interesting and topical. I have the following questions:

1. What is the main incentive (reason) to use a mobile phone for the entire process?

    I admit, I expected the use of photogrammetry using a high-quality camera (digital SLR) and some of the SW. Not that I underestimate cell phone cameras.

2. Did the study consider (will consider) the comparison of different smartphones for use in the given issue?

3. Does choosing another high-quality smartphone affect the overall result? (personally, I would be interested in the top-of-the-line Huawei brand, but also Apple or other comparable optics quality).

4. Chapter 2 describes the basic features of individual SW products. However, I would be more interested in the reason for the choice. For example, why Agisoft metashape and not PhotoModelerScaner, etc.

5. Would it be possible to include some financial data in the article (in which price ranges did the authors move during their research, sw price, ,,,)?

I find the article very valuable and I thank the authors and congratulate them on a great job

Reviewer 3 Report

Comments and Suggestions for Authors

The manuscript is useful and interesting. Nevertheless, the paper requires to go through a careful revision before publication. I invite the authors to modify their manuscript according to the below comments.

The introduction does not sufficiently justify the need for smartphone-based photogrammetry and its potential advantages over other methods. More comparative analysis with existing techniques, such as laser scanners or structured light scanners, would strengthen the rationale for this study. In addition, other fields using smartphone-based photogrammetry are suggested (https://doi.org/10.3390/rs12111889 ; https://doi.org/10.1016/j.enggeo.2023.107170 ; https://www.mdpi.com/2072-4292/14/20/5187 ).

The methods section lacks detailed information on the selection criteria for the volunteers and head mannequins. It is important to clarify how representative the sample is and whether any specific characteristics were considered.

The results section provides numerical values for the obtained 3D meshes and linear measurements but does not adequately discuss the significance of these findings. It would be beneficial to include statistical analyses and compare the results with previous studies to evaluate the accuracy and reliability of the proposed method.

The discussion section does not address potential limitations or challenges encountered during the study. It is crucial to acknowledge any limitations or sources of error that may have influenced the results and discuss how they could be addressed in future research.

The authors mention the use of Agisoft Metashape and PhotoMeDAS software for photogrammetric processing, but no information is provided on the specific settings or parameters used. Including these details would enhance the reproducibility of the study and allow other researchers to replicate the experiments.

The manuscript lacks visual aids such as figures or diagrams to illustrate the process of capturing images and generating 3D models. Adding visual representations would help readers better understand the methodology and results.

The conclusion section does not provide a clear summary of the main findings and their implications. It should emphasize the novel contributions of the study and highlight the potential applications of smartphone-based photogrammetry in the medical field.

The language and writing style of the manuscript need improvement. There are several grammatical errors and awkward sentence structures throughout the text. A thorough proofreading and editing process would enhance the overall clarity and readability of the manuscript.

Round 2

Reviewer 1 Report

Comments and Suggestions for Authors

It seems the authors address my concerns to some extent. I suggest the author further discuss and compare the importance of the markers, i.e., how the results will be without markers.

Reviewer 3 Report

Comments and Suggestions for Authors

The authors have carefully revised the manuscript and addressed all the issues raised by me. It is an interesting work. This version can be accepted for publication.
